

# On the derivation of zonal and meridional wind components from Aeolus horizontal line-of-sight wind

Isabell Krisch[1], Neil P. Hindley[2], Oliver Reitebuch[1], and Corwin J. Wright[2]

[1]Deutsches Zentrum für Luft- und Raumfahrt e.V. (DLR), Institut für Physik der Atmosphäre, Oberpfaffenhofen, Germany
[2]Centre for Space, Atmospheric and Oceanic Science, University of Bath, UK

*Correspondence to*: Isabell Krisch (isabell.krisch@dlr.de)

**Abstract.** Since its launch in 2018, the European Space Agency's Earth Explorer satellite Aeolus has provided global height resolved measurements of horizontal wind in the troposphere and lower stratosphere for the first time. Novel datasets such as these provide an unprecedented opportunity for the research of atmospheric dynamics and provide new insights into the dynamics of the upper troposphere and lower stratosphere (UTLS) region. Aeolus measures the wind component along its horizontal line-of-sight, but for the analysis and interpretation of atmospheric dynamics, zonal and/or meridional wind components are most useful. In this paper, we introduce and compare three different methods to derive zonal and meridional wind components from the Aeolus wind measurements. We find that the most promising method involves combining Aeolus measurements during ascending and descending orbits. Using this method, we derive global estimates of the zonal wind in the latitude range 79.7° S to 84.5° N with errors of less than 5 ms$^{-1}$ (at the 2-sigma level). Due to the orbit geometry of Aeolus, the estimation of meridional wind in the tropics and at midlatitudes is more challenging and the quality is less reliable. However, we find that it is possible to derive meridional winds poleward of 70° latitude with absolute errors typically below ±5 ms$^{-1}$ (at the 2-sigma level). This further demonstrate the value of Aeolus wind measurements for applications in weather and climate research, in addition to their important role in numerical weather prediction.

## 1 Introduction

In August 2018, the European Space Agency (ESA) launched the Earth Explorer Satellite Aeolus (ESA, 2008; Reitebuch et al., 2020). It carries the first wind lidar in space – the Atmospheric LAser Doppler Instrument (ALADIN; e.g. Reitebuch, 2012a), which provides global profiles of line-of-sight (LOS) winds and optical properties of clouds and aerosols (Stoffelen et al., 2020; Flament et al., 2021). ALADIN is a high spectral resolution Doppler wind lidar which is operated at a wavelength of 354.8 nm. With its high-power laser, ALADIN can penetrate the atmosphere and acquire measurements from roughly 30 km altitude down to either the ground or to the highest optically thick cloud layer.

The main objective of the Aeolus mission is to improve numerical weather prediction (NWP; ESA, 1999; Andersson, 2018). Multiple NWP centres have already shown the positive impact of Aeolus data (e.g. Rennie et al., 2021) and started its operational assimilation. However, detailed wind information is not only beneficial for NWP, but also for atmospheric



dynamics research (ESA, 2008; Reitebuch, 2012b; Stoffelen et al., 2020). Many dynamic features show characteristic wind patterns and/or are strongly influenced by the prevailing wind. Especially in the upper troposphere / lower stratosphere (UTLS) region, Aeolus measurements can provide valuable information for the investigation of dynamic features such as gravity waves (GWs; e.g. Banyard et al., 2021a), Kelvin waves (e.g. Zagar et al., 2021), sudden stratospheric warmings (SSWs; e.g. Wright et al., 2021), or the quasi-biennial oscillation (QBO; e.g. Banyard et al., 2021b).

Most of these dynamic features show very characteristic patterns along one of the three wind directions (zonal, meridional or vertical). Thus, a separation of the LOS wind as measured by Aeolus into these three wind components – zonal, meridional and vertical – could significantly improve the usefulness of Aeolus data for atmospheric dynamics research. This paper will investigate how the Aeolus LOS wind can be used to derive zonal and meridional wind components, and which limitations apply. The vertical wind component will not be considered here, because the typical horizontal and vertical averaging lengths

of Aeolus measurements are too coarse and the error estimates too high to properly resolve even strong vertical winds, e.g. in the atmospheric boundary layer, within convection, or in gravity waves.

For the assimilation of Aeolus data in NWP, such a conversion of Aeolus LOS wind to zonal, meridional and vertical wind component is not necessary, as the model forward operator includes a projection of the respective model winds onto the Aeolus horizontal line-of-sight (HLOS). However, Lux et al. (2020) experienced a similar conversion issue when comparing Aeolus

HLOS wind with wind measurements from the ALADIN Airborne Demonstrator (A2D). As both instruments were pointing in slightly different azimuth directions, the measured wind components from satellite and airborne instrument were not directly comparable to each other and a conversion was necessary. Lux et al. (2020) used model data from the European Centre for Medium-Range Weather Forecasts (ECMWF), which is contained in the Aeolus Level 2C (L2C) product, to correct for the wind speed differences originating from the different viewing geometries.

The present paper will investigate if and how a conversion to zonal and meridional wind is possible using solely Aeolus measurements without requiring additional information (for example from NWP models). The presented methods are especially useful for studying atmospheric phenomena over a long time periods and large geographic areas (e.g. zonal mean values), where larger Aeolus observation samples can be used. The aim of this paper is to provide scientists studying synoptic scale phenomena especially in the stratosphere (e.g. SSWs, QBO) with a toolbox to convert the Aeolus L2B products into

zonal and meridional wind components keeping the limitations of such conversion methods in mind.

Section 2 describes the Aeolus measurement geometry, Sections 3 and 4 describe three different techniques to estimate zonal and meridional wind components and the expected errors of these methods from theory. Section 5 briefly describes the colocation of ascending and descending measurements, which is required for one of the methods, and what impact this can have on the derived winds. In Section 6, errors in the methods are derived statistically using a simulated dataset based on

reanalysis data. In Section 7 the results are summarised and discussed.





## 2 Aeolus measurement principle

The Aeolus satellite orbits the Earth at an altitude of 320 km on a polar, sun-synchronous, dawn-dusk orbit at an inclination of 97° and a repeat cycle of 7 days (ESA, 2021a). Aeolus' only payload, ALADIN, points a large telescope at an angle of 35° off-nadir perpendicular to the satellite flight direction and towards the dark side of the terminator. Laser pulses are emitted in this direction at a frequency of 50.5 Hz and the return signal from Rayleigh scattering at molecules in the atmosphere and from Mie scattering at particles (e.g. clouds and aerosol) is measured at high temporal and spectral resolution. The high spectral resolution is necessary to provide an accurate estimation of the Doppler shift and, thus, the wind velocity in the atmosphere. The high temporal resolution makes it possible to vertically sample the atmosphere in altitude bins as small as 250 m. Vertical sampling is achieved through ranging with the time-of-flight principle. Therefore, the altitude bins are often called range bins. To achieve a reasonable signal-to-noise ratio (SNR), an accumulation of the return signal of multiple laser pulses is done on-board at detector level. Such an accumulation is only possible for 24 range bins simultaneously. The range bin thicknesses and top heights can be adjusted on-board to the needs of numerical weather prediction (NWP) and science applications. They vary along orbit and from season to season to enhance, for example, the number of available measurements close to strong wind gradients, e.g. in the vicinity of jet streams. Usually, the range bin thicknesses are between 500 m and 2 km and the top-most measurement altitude between 17.5 km and 25 km (ESA, 2021b).

On top of this on-board accumulation of multiple lidar pulses, the signal is also averaged over multiple measurements within the ground processing chain to further improve the SNR and reduce the random errors (Reitebuch et al., 2018; Rennie et al., 2020). The horizontal averaging length is usually on the order of 90 km along track under clear sky conditions (Rayleigh wind product) and can reach lengths as short as to 3-12 km where aerosol and clouds are present and increase the backscatter signal (Mie wind product).

ALADIN measures the frequency shift $\Delta f$ between the outgoing laser signal (with frequency $f_0$) and the return signal. This frequency shift is related to the wind velocity along line-of-sight (LOS), $w_{LOS}$, through the Doppler-Equation:

$$\Delta f = 2 f_0 \frac{w_{LOS}}{c}, \tag{1}$$

with $c$ the speed of light.

The LOS wind $w_{LOS}$ contains contributions from both the horizontal and the vertical wind (see Fig. 1 a):

$$w_{LOS} = w_{HLOS} \cos\alpha + w_v \sin\alpha, \tag{2}$$

where $\alpha$ is the elevation of the satellite-to-target pointing vector, $w_{HLOS}$ is the horizontal projection of the LOS wind (HLOS), and $w_v$ is the vertical wind. Due to the earth's curvature, the elevation of the satellite-to-target pointing vector $\alpha$ varies depending on measurement altitude and geolocation between 52° and 54°.

The wind $w_{HLOS}$ consists of a meridional and a zonal component (see Fig. 1 b & c):

$$w_{HLOS} = w_h \cos(\theta - \varphi) = -u \sin\theta - v \cos\theta, \tag{3}$$



where $\theta$ is the topocentric[1] azimuth of the target-to-satellite pointing vector measured clockwise from north, $\varphi$ is the topocentric azimuth of the horizontal wind vector measured clockwise from north, and $u$ and $v$ are the zonal and meridional winds, respectively.

Both the elevation angle $\alpha$ and the azimuth angle $\theta$ are reported in the Aeolus Level 2B (L2B) product (De Kloe et al., 2020; Rennie et al. 2020). The standard L2B wind product of Aeolus is the HLOS wind $w_{HLOS}$.

In the Aeolus measurement regime (vertical resolution of 500 m – 2 km, horizontal resolution of 3-90 km), the vertical wind $w_v$ is usually much smaller than the horizontal wind $w_{HLOS}$. Therefore, the vertical wind $w_v$ is assumed to be zero in the Aeolus processing chain for the L2B product and the measured LOS wind $w_{LOS}$ is converted into a HLOS wind $w_{HLOS}$ as follows:

$$w_{HLOS} = \frac{w_{LOS}}{\cos \alpha}. \tag{4}$$

The error in HLOS wind due to this simple assumption is

$$\Delta w_{HLOS} = w_v \frac{\cos \alpha}{\sin \alpha} \approx 1.32 \cdot w_v. \tag{5}$$

Schumann (2019) suggests that, at horizontal scales below ~50 km, horizontal and vertical velocity can, under certain

circumstances, reach a similar order of magnitude. Above these scales, the horizontal motion usually is much stronger than the vertical motion and $\Delta w_{HLOS}$ becomes negligible. Thus, for the Rayleigh wind product (horizontal resolution of 90 km), Eq. (4) should provide reliable estimates of $w_{HLOS}$. Under certain atmospheric conditions (e.g. strong convection, gravity waves), however, this assumption might cause significant errors in the determined Mie wind product (horizontal resolution of 3-12 km).

**3 Methods to estimate zonal and meridional wind components from Aeolus horizontal line-of-sight wind**

In this section, we briefly describe the three methods to convert Aeolus HLOS wind $w_{HLOS}$ into zonal and meridional component that are assessed in this paper. A detailed discussion about the validity of assumptions and simplifications behind each method and an in-depth error analysis will follow in the next sections.

**3.1  Method 1: projection of HLOS wind**

The first method to convert Aeolus $w_{HLOS}$ into zonal and meridional wind components is a simple projection of $w_{HLOS}$ onto the zonal and meridional axis direction:

$$u_1^* = -w_{HLOS} \sin \theta, \tag{6}$$
$$v_1^* = -w_{HLOS} \cos \theta. \tag{7}$$

The geometric concept of this method is shown in Fig. 2a. This conversion is done for each measurement of Aeolus $w_{HLOS}$ separately.

---

[1] The origin of a topocentric coordinate system is a point on the Earth's surface. All horizontal coordinates in reference frames parallel to the Earth's surface are topocentric.





## 3.2 Method 2: assumption of zero meridional (zonal) wind

The second method uses the assumption of zero meridional (zonal) wind for determining the zonal (meridional) wind component:

$$u_2^* = -\frac{w_{HLOS}}{\sin\theta}, \tag{8}$$

$$v_2^* = -\frac{w_{HLOS}}{\cos\theta}. \tag{9}$$

Again, the geometric concept of this method is shown in Fig. 2b and the conversion can be done for each measurement of Aeolus $w_{HLOS}$ separately.

## 3.3 Method 3: use of ascending and descending orbits

The third method uses the geometrical differences between ascending and descending orbits ($\theta_{asc} - 360° = -\theta_{dsc}$) and combines measurements from both to more accurately estimate the true zonal and meridional wind over a specific region. To derive the zonal wind, we use:

$$
\begin{aligned}
u_3^* &= -0.5 \cdot \left(\frac{w_{HLOS,asc}}{\sin\theta_{asc}} + \frac{w_{HLOS,dsc}}{\sin\theta_{dsc}}\right) \\
&= -0.5 \cdot \left(\frac{-u_{asc}\sin\theta_{asc} - v_{asc}\cos\theta_{asc}}{\sin\theta_{asc}} + \frac{-u_{dsc}\sin\theta_{dsc} - v_{dsc}\cos\theta_{dsc}}{\sin\theta_{dsc}}\right) \\
&= 0.5 \cdot (u_{asc} + u_{dsc}) + 0.5 \cdot (v_{asc}\cot\theta_{asc} + v_{dsc}\cot\theta_{dsc}) \\
&= 0.5 \cdot (u_{asc} + u_{dsc}) + 0.5 \cdot \cot\theta_{asc} \cdot (v_{asc} - v_{dsc}),
\end{aligned}
\tag{10}
$$

using $\cot\theta_{asc} = -\cot\theta_{dsc}$. Under perfect conditions ($v_{asc} = v_{dsc}$), the meridional wind component cancels out and only the zonal wind component remains. This is also shown in Fig. 2c.

The meridional wind component can be calculated in a similar manner:

$$
\begin{aligned}
v_3^* &= -\frac{w_{HLOS,asc} + w_{HLOS,dsc}}{\cos\theta_{asc} + \cos\theta_{dsc}} \\
&= -\frac{-u_{asc}\sin\theta_{asc} - v_{asc}\cos\theta_{asc} - u_{dsc}\sin\theta_{dsc} - v_{dsc}\cos\theta_{dsc}}{\cos\theta_{asc} + \cos\theta_{dsc}} \\
&= \frac{u_{asc}\sin\theta_{asc} + v_{asc}\cos\theta_{asc} - u_{dsc}\sin\theta_{asc} + v_{dsc}\cos\theta_{asc}}{2 \cdot \cos\theta_{asc}} \\
&= 0.5 \cdot \tan\theta_{asc} \cdot (u_{asc} - u_{dsc}) + 0.5 \cdot (v_{asc} + v_{dsc}).
\end{aligned}
\tag{11}
$$

Here, $\sin\theta_{asc} = -\sin\theta_{dsc}$ and $\cos\theta_{asc} = \cos\theta_{dsc}$ are used. As for the zonal wind calculation, under perfect conditions ($u_{asc} = u_{dsc}$), now the zonal wind component cancels out and only the meridional wind component remains.

In contrast to the two methods introduced before, the results of this method always are a temporal and spatial average as multiple measurements acquired at different times and locations are combined.



## 4 Theoretical estimation of systematic errors

The systematic errors of all three methods are correlated to trigonometric functions of the azimuth angle $\theta$. All relevant trigonometric functions of $\theta$ are depicted in Fig. 3 with respect to Aeolus measurement latitude.

### 4.1 Error of Method 1

For Method 1, the systematic error in the estimated zonal (meridional) wind can be calculated as follows:

$$\varepsilon_{u_1^*} = -w_{HLOS}\sin\theta - u = -w_h\cos(\theta - \varphi)\sin\theta + w_h\sin\varphi = w_h(\sin\varphi - \cos(\theta - \varphi)\sin\theta)$$
$$= -w_h\cos\theta\sin(\theta - \varphi)\,, \tag{12}$$

$$\varepsilon_{v_1^*} = -w_{HLOS}\cos\theta - v = -w_h\cos(\theta - \varphi)\cos\theta + w_h\cos\varphi = w_h(\cos\varphi - \cos(\theta - \varphi)\cos\theta)$$
$$= w_h\sin\theta\sin(\theta - \varphi)\,. \tag{13}$$

Both errors are proportional to (i) the true horizontal wind $w_h$, (ii) the sine of the angle difference between HLOS and the horizontal wind direction, and (iii) the sine or cosine of the azimuth angle. Thus, if the horizontal wind direction is very close to HLOS, the errors become small. Additionally, if HLOS is almost east-west oriented ($\theta = 90°$ or $\theta = 270°$, close to the equator), the contribution of the meridional wind to $w_{HLOS}$ becomes very small and thus the error in the derived zonal wind is also reduced, while the error in the derived meridional wind becomes very large. For HLOS almost oriented north-south ($\theta = 0°$ or $\theta = 180°$, close to the poles), the inverse situation applies: the contribution of the zonal wind to $w_{HLOS}$ becomes very small, the error in the derived meridional wind gets small, and the error in the derived zonal wind becomes large. If the true horizontal wind $w_h$ points in the direction of HLOS ($\theta = \varphi$), both systematic errors vanish.

Additionally, if HLOS is oriented exactly along the zonal axis, the zonal wind error $\varepsilon_{u_{HLOS,1}}$ vanishes. Near the equator, $\theta$ closely approaches $270°$ ($90°$) for ascending (descending) orbits. Thus, the zonal wind error does not completely vanish close to the equator, but does become small. In the same way, the meridional wind error vanishes for an HLOS oriented along the meridional axis. This happens over the poles when switching from an ascending to a descending orbit.

### 4.2 Error of Method 2

For Method 2, the systematic error in the estimated zonal (meridional) wind is directly related to the true meridional (zonal) wind as:

$$\varepsilon_{u_2^*} = -\frac{w_{HLOS}}{\sin\theta} - u = \frac{u\sin\theta + v\cos\theta}{\sin\theta} - u = v\cot\theta\,, \tag{14}$$

$$\varepsilon_{v_2^*} = -\frac{w_{HLOS}}{\cos\theta} - v = \frac{u\sin\theta + v\cos\theta}{\cos\theta} - v = u\tan\theta. \tag{15}$$

Here, the error is related to the tangent or cotangent of the azimuth angle. Thus, if the HLOS direction is more than $45°$ away from the zonal axis, the error in the zonal wind component rapidly increases. Similarly, for HLOS directions more than $45°$ away from the meridional axis, the meridional wind errors become very large. This means that Method 2 only yields reasonable results for the zonal wind close to the equator ($<70°$ latitude) and for the meridional wind close to the poles ($>70°$ latitude).





An additional point worth mentioning is that both the tangent and the cotangent function are point symmetric around $n \cdot \frac{\pi}{2}$ ($n \in \mathbb{Z}$). This means that the errors for ascending and descending orbits have opposite signs. We take advantage of this fact in Method 3.

### 4.3 Error of Method 3

Method 3 is based on the combination of at least two measurements. Thus, we need to take the differences in zonal wind $\Delta u$ and meridional wind $\Delta v$ between these two or more measurements on ascending and descending paths into account for the error analysis:

$$\varepsilon_{u_3^*} = 0.5 \cdot (u_{asc} + u_{dsc}) + 0.5 \cdot \cot \theta_{asc} \cdot (v_{asc} - v_{dsc}) - \bar{u} = 0.5 \cdot \cot \theta_{asc} \cdot \Delta v, \tag{16}$$

$$\varepsilon_{v_3^*} = 0.5 \cdot \tan \theta_{asc} \cdot (u_{asc} - u_{dsc}) + 0.5 \cdot (v_{asc} + v_{dsc}) - \bar{v} = 0.5 \cdot \tan \theta_{asc} \cdot \Delta u. \tag{17}$$

Here, the theoretical error of one wind component is correlated with the difference in the other wind component multiplied by either $0.5 \cdot \cot \theta_{asc}$ (zonal wind error) or $0.5 \cdot \tan \theta_{asc}$. In general, the same is valid here as for method 2: For HLOS directions oriented almost east-west, the cotangent is low and the tangent approaches infinity; for HLOS directions oriented almost north-south, the tangent is low and the cotangent approaches infinity. However, due to the factor of 0.5, the angles for which a total factor of one is reached are $\pm 63.4°$ with respect to the optimal direction, i.e. east-west for zonal and north-south for meridional

wind derivation.

With only one Aeolus satellite in orbit, there are never two measurements (one on ascending and one on descending path) at the exact same time and location. Method 3 therefore relies on temporal and spatial interpolation, described in more details in Section 5. Thus, in addition to the theoretical errors in Eq. (16) and (17), interpolation errors also need to be considered for the results of Method 3. These interpolation errors are analysed and discussed in more detail in Section 6.3 using reanalysis data.

### 5 Colocation of ascending and descending measurements

The primary challenge in applying the third method is to find suitable pairs of ascending and descending measurement. One suitable approach, which is used in this paper, is based on simple binning in latitude and linear interpolation between nearest neighbours in longitude and time.

For the analysis in this paper, all Aeolus measurements are first interpolated to a constant altitude and then binned into latitude

bins of 1° width ($\pm 0.5°$ around each full degree latitude). Next, for each individual measurement, the nearest neighbours in longitude (one east and one west) but with different orbit phase (ascending vs. descending) are identified within the time period from twenty hours before to twenty hours after the measurement time. In this way, four nearest neighbours are determined: the earlier west neighbour (EWN), the earlier east neighbour (EEN), the later west neighbour (LWN), and the later east neighbour (LEN).





The two early and two late nearest neighbours are then linearly interpolated in space onto the original measurement location
     and afterwards interpolated in time. This interpolation is done for both the HLOS wind $w_{HLOS}$ and the azimuth angle $\theta$. In this
     way, a corresponding ascending (descending) measurement point is constructed for each descending (ascending) Aeolus
     measurement.

     Figure 4 shows statistically, how far away the nearest neighbours are from the original measurement location. More than 94%
of the nearest neighbours are less than 22.7° (~2500 km at the equator) away in zonal direction and have a temporal distance
     of less than 15.5 h. Most of them (~70%) are actually acquired between 10 h and 14 h earlier or later. Both the turning point
     in the longitudinal distance as well as the step-wise behaviour of the temporal distance, which are observed in Figure 4, have
     their origin in the Aeolus orbit geometry: one orbit of Aeolus takes roughly 1.5 h and two consecutive orbits are shifted by
     ~20° in longitude.

The validity of this colocation methods and its influence on the error statistics will be discussed in detail in Section 6.3.

## 6 Error assessment using simulated Aeolus observations from ERA5

To assess the errors in the methods statistically, Aeolus-like HLOS wind measurements are constructed by sampling fifth
generation European Centre for Medium-Range Weather Forecasts (ECMWF) reanalysis data (ERA5; Hersbach et al., 2018)
using real Aeolus Rayleigh clear measurement locations (time, longitude, latitude, altitude and azimuth angle) for January
2021. For simplicity and comparability of the different methods, these real Aeolus measurement locations are interpolated in
our study to a constant altitude of 15km. ERA5 hourly zonal and meridional wind data on a 1°x1° grid are then trilinearly
interpolated in time (1D) and space (only horizontally, thus 2D) onto the Aeolus measurement locations. At each Aeolus
measurement location, the simulated horizontal-line-of-sight wind $w_{HLOS}$ is calculated from the interpolated ERA5 winds $u$
and $v$, and the Aeolus topocentric azimuth angle $\theta$ following Eq. (3). Figure 5c shows an example of $w_{HLOS}$ for one day.

These simulated Aeolus measurements are used to determine zonal and meridional wind components following the three
methods described above. We then compare these estimates to the original, interpolated ERA5 winds to derive statistical error
estimates. Absolute errors of all three methods are shown as examples for the Aeolus tracks on 15 January 2021 in Figure 6.
Statistical distributions of the absolute and relative errors of all the single wind estimates for the whole month of January 2021
are shown in Figure 7. The results are discussed in the following for zonal and meridional wind separately.

## 6.1 Quality of zonal wind estimation

In summary, all three methods produce reliable zonal wind estimates between 70° S and 70° N with absolute errors typically
below 5 ms$^{-1}$. At a first glance, 5 ms$^{-1}$ might seem high. However, one should keep in mind that the typical random error of
Aeolus HLOS wind measurements in the UTLS is between 3 and 7 ms$^{-1}$ (Rennie et al., 2021) and, thus, in a similar range.
Poleward of 70° latitude, the zonal wind errors of Method 1 & 2 increase strongly, whereas the errors of Method 3 remain at
a low level, except for the very northernmost and southernmost portions of each orbit. The zonal wind errors of Method 3





remain below 5 ms$^{-1}$ at the 2-sigma level as long as $|\theta_{Aeolus} - 180°| > 30°$; for the Aeolus orbit, this is the case for all latitudes between 79.7° S and 84.5° N.

In Figure 7a (Method 1), a latitude-dependent bias is clearly visible as an offset between the centre of the distribution and the zero-difference axis. Methods 2 & 3 (Figure 7 b & c) do not show this offset, indicating that such a bias is absent from these
methods. The widths of the error distributions do however vary for all three methods with latitude. Method 1 also shows an increase of the mean relative error close to the poles (Figure 7 d), suggesting that this method underestimates the zonal wind in this region.

## 6.2 Quality of meridional wind estimation

Of the three methods that we describe here, Method 3 is the only method able to produce reliable meridional winds at all
latitudes. Especially poleward of 70° latitude the derivation of the meridional wind is reasonable with absolute errors usually below ±5 ms$^{-1}$ (at the 2-sigma level). Between 70° S and 70° N however, errors increase and often approach ±30 ms$^{-1}$ at the 2-sigma level or ±15 ms$^{-1}$ at the 1-sigma level. Both mean absolute and mean relative error show a small dependency on latitude (Figure 7 i & l).

The absolute errors in the meridional wind derived using Method 1 (Figure 6b) appear to be strongly correlated with the true
meridional wind. This is confirmed by the relative errors in Figure 7j, which are close to unity at most latitudes. Thus, Method 1 cannot derive the correct magnitude of the meridional wind.

For Method 2 (Figure 6d), meridional wind errors increase dramatically everywhere except very close to the poles and, even at these high latitude-locations, some regions (e.g. north of Russia) exhibit very high absolute errors. Both absolute and relative meridional wind errors (Figure 7 h & k) confirm this finding.
It is important to note that the Aeolus measurement geometry is significantly more favourable for the derivation of zonal than meridional winds and that, regardless of the data treatment used, meridional wind estimates might contain large errors, especially far away from the poles.

## 6.3 Influence of temporal and spatial interpolation on the accuracy of Method 3

As mentioned above, Method 3 relies on the temporal and spatial interpolation of measurement data taken at different locations
and at different points in time. This interpolation is the main error source of this method and its influence on accuracy will be assessed in this section. For this, the set of collocated measurements (earlier west neighbour – EWN, earlier east neighbour – EEN, later west neighbour – LWN, and later east neighbour – LEN) described in Section 5 is used, but populated this time with ERA5 model data having either the same time or the same location as the centre measurement. These constant time or constant location neighbours are then used exactly as before to derive zonal and meridional wind.
Figure 8 shows the absolute errors for zonal and meridional wind for the standard Method 3 (a & d, same as Figure 7 c & i) and for constant time (b & e) and constant location (c & f). The errors due to both temporal as well as spatial interpolation strongly depend on the distance to the nearest neighbour: The closer in time (Panel h) or space (Panel g) the nearest neighbour





is, the smaller is the error. This is clearly visible especially for spatial interpolation: when the distance to the nearest neighbour is close to 0° (e.g. around 55° S, 20° S and 40° N), the errors are relatively small, whereas when the nearest neighbour is far,

the errors are relatively large. A similar behaviour is observed for the temporal interpolation: when the nearest neighbour is closer in time (e.g. south of 60° S and north of 70° N), the errors drop significantly.

In addition to this dependence on the distance to the nearest neighbours, the absolute errors are also correlated with the variability of the wind speeds (Panels i & j). Close to the jet stream regions (at around 50° S and 40° N), the errors are enhanced. This can be seen more clearly for the temporal interpolation, because here the effect of the distance to the nearest neighbour is

less variable. However, it can also be observed for the spatial interpolation, where e.g. the error minimum at 40° N is less pronounced as the error minimum at 20° S even though the longitudinal distance is the same.

The total errors of Method 3 (Panels a & d) are a mixture of both temporal and spatial interpolation. As expected, the two are not independent and, thus, the total error is smaller than the pure summation of both errors. However, characteristic features of both temporal and spatial interpolation can be observed in the total error statistic. At 40° S for example, the largest errors

are due to a combination of large spatial distances between the measurements and high temporal variability of the wind speeds in the jet stream region. At 20° S, the errors decrease due to the small spatial distance between nearest neighbours. At 40°N, on the other hand, the total errors are less prominent, because the small spatial distances between collocated measurements balance the large errors due to temporal variability of the atmosphere.

## 6.4 Calculating daily and zonal mean of zonal wind with all presented methods

One application of Aeolus data in the stratosphere is the analysis of the QBO signal. This signal is usually analysed and monitored by looking at daily zonal means of the zonal wind. This section will briefly discuss the accuracy of estimating such daily zonal means of the zonal wind from Aeolus measurements using the three conversion methods introduced before.

Here, the daily zonal means are calculated in latitude bins of width $\pm 2.5°$ whose bin centres are stepped in latitude by 5°. They are first estimated from hourly ERA5 model data on the original 1° longitude by 1° latitude grid. The estimates are then

compared to daily zonal means from ERA5 data sampled on real Aeolus measurement locations. The difference, which is a measure of the sampling bias of Aeolus, is shown in Figure 9 a. The enhancement of the sampling bias close to 40° N is related to missing wind data on certain orbits due to calibration measurements. Due to these calibration measurements, the zonal wind structure is not sampled equidistantly anymore, which leads to a weekly alternating bias of up to 3 ms$^{-1}$. Using Kalman filtering or spectral reconstruction instead of simple binning and averaging could reduce this sampling bias in future applications.

Panels b, c & d of Figure 9 show the errors in the daily zonal mean estimates for the three Methods introduced before. The sampling bias of Aeolus (Panel a) has been removed here. Close to the equator all methods produce reasonable zonal mean zonal wind estimates with errors below 0.5 ms$^{-1}$. However, poleward of 20° latitude the estimates of Method 1 strongly degrade and systematic biases of several ms$^{-1}$ are observed.

As already discussed in Section 4.2, the errors in $u_2^*$ of ascending and descending orbits have opposite signs. Thus, they cancel

against each other when averaging in zonal direction. Hence, the daily zonal means of zonal wind calculated with Method 2



(Panel b) overall have very low errors typically below 0.5 ms$^{-1}$ except for the most northern and southern latitude band. Here, the geometric projection of $w_{HLOS}$ to $u_2^*$ for points very close to the orbit turning point introduces large errors, which contaminate the whole latitude band.

The errors of Method 3 are very similar to the ones of Method 2. However, the temporal and spatial interpolation between
collocated measurements introduces some small additional errors.

For the calculation of daily zonal mean winds, one alternative to the complex colocation and interpolation introduced in Section 5 is to solely bin all $w_{HLOS}$ measurements within 360° longitude and 24 h separately for ascending and descending orbits and then apply Method 3 on these averaged values. This is nothing other than reversing the order of mathematical operations of Method 2 above, i.e. by first averaging and then projecting geometrically. The results of this method are shown
in Panel (e). The obtained values are exactly the same as for Method 2 except for the most  northern and southern latitude bands. By reversing the order of the mathematical operations, the influence of single measurements very close to the turning point decreases and the daily zonal mean zonal wind estimates close to the poles become more reliable.

Thus, for calculating large scale averages (both in time and/or space) of zonal and meridional wind, it is recommended to first perform the averaging on ascending and descending $w_{HLOS}$ measurements separately and then combine both with Method 3 to
determine the zonal and/or meridional wind component.

**7 Summary, Discussion and Conclusions**

In this paper, three different methods for the estimation of zonal and meridional wind components from Aeolus HLOS measurements are compared and their systematic errors are investigated. This is done first analytically and then by using simulated measurements based on ERA5 reanalysis data.
In general, the quality of the estimation strongly depends on the angle between Aeolus HLOS and the different cardinal axes. If the HLOS direction is almost zonal, the zonal wind component can be derived very well. This is the case equatorward of roughly 60° latitude, where the HLOS direction is only 10° to 15° (-10° to -15°) off the zonal direction for ascending (descending) orbits. Similarly, the derivation of meridional wind works well close to the poles, where HLOS gets closer to the meridional direction.
For Method 1, which simply projects the Aeolus HLOS wind onto the zonal and meridional axes, all errors vanish if the true horizontal wind is in line with the HLOS. However, this is rarely the case. Overall this method produces reasonable zonal winds equatorward of 70° latitude, but also introduces small latitudinal biases. The meridional winds of Method 1 show everywhere large random deviations and overall represent a strong underestimation.

Method 2, which relies on the assumption of zero meridional (zonal) wind to derive the zonal (meridional) wind component,
shows a similar behaviour as Method 1 for the zonal wind accuracy. However, it does not show any latitudinal variation of the mean bias. The meridional wind derived with Method 2 cannot be recommended for use in scientific analyses.





Method 3, which utilizes the geometrical differences between ascending and descending orbits, produces the best results. Both wind components have reasonable absolute and relative error characteristics and do not show any sign of systematic underestimation. The derived zonal winds show enhanced errors only very close to the poles (poleward of 84.5° N and 79.7° S respectively). Due to the Aeolus measurement geometry, with HLOS usually oriented closer to the zonal than the meridional direction, the meridional wind errors are much larger (by a factor of 4) than the zonal wind errors, except poleward of 70° latitude.

For the reconstruction of zonal winds, all three methods can generally be used within the latitude band 70° S to 70° N, with varying degrees of error. Method 2 & 3 have the advantage that they do not introduce latitudinal mean biases. Poleward of 70° latitude, however, only Method 3 is recommended for use.

For the reconstruction of meridional winds, Method 3 is the only method able to produce reasonable results without unacceptably large errors (Method 2) and/or significant underestimation (Method 1). However, one should always keep in mind that the Aeolus measurement geometry is generally more favourable for deriving the zonal wind component in the tropics and midlatitudes and that meridional wind estimates in this region have to be treated with care.

The largest disadvantage of Method 3, as applied in this paper, is that it combines measurements in a time period of around ±14 h (up to ±20 h) and a distance of up to 2500 km (20° longitude) around the original measurement location. Due to this, the errors of this method are strongly correlated with the spatial and temporal distance between combined measurements and the spatial and temporal variability of the atmosphere. More complex grouping algorithms, using for example spectral reconstruction or Kalman filtering, might help to reduce these errors and should be investigated in the future.

Due to this combination of temporally and spatially distant measurements, this method is only suitable to derive large scale wind structures that vary relatively slowly in time. However, the characteristics of many dynamic features in the UTLS (e.g. the wind reversal of an SSW, or the large-scale descending structure of the QBO) are typically derived from large-scale averages in time (e.g. daily/monthly averages; Baldwin et al., 2001; Butler et al., 2017) and/or longitude (e.g. zonal mean; Baldwin et al., 2001; Baldwin et al., 2021). Thus, for the analysis of such large-scale phenomena, Method 3 is a very good way to gain information on both meridional and zonal wind from Aeolus measurements.

**Data availability**

ERA5 data (Hersbach et al., 2018) were downloaded from the Copernicus Climate Change Service (C3S) Climate Data Store (CDS). The results contain modified Copernicus Climate Change Service information 2021. Neither the European Commission nor ECMWF is responsible for any use that may be made of the Copernicus information or data it contains.

Aeolus Baseline 11 data were obtained from the ESA Aeolus Online Dissemination System (https://aeolus-ds.eo.esa.int/oads/access/). The processor development, improvement and product reprocessing preparation are performed by the Aeolus DISC (Data, Innovation and Science Cluster), which involves DLR, DoRIT, ECMWF, KNMI, CNRS, S&T, ABB and Serco, in close cooperation with the Aeolus PDGS (Payload Data Ground Segment).



**Author contribution**

All authors contributed to the selection/invention of the used wind conversion methods. IK performed the data analysis. OR provided support with respect to Aeolus specific issues. NH provided advise on the grouping of ascending and descending orbits. All authors contributed to the paper preparation and the interpretation of the results.

**Competing interests**

The authors declare that they have no conflict of interest.

**Acknowledgements**

The authors thank Tim Banyard and Norman Wildman for their review of the manuscript and the very helpful comments.

**Financial support**

This research has been supported by the European Space Agency in the frame of the Aeolus DISC (grant no. 4000126336/18/I-BG), and the Bundesministerium für Bildung und Forschung in the frame of the ROMIC-II / QUBICC project (grant no. 01LG1905D). Corwin Wright is funded by Royal Society University Research Fellowship UF160545, and Corwin Wright and Neil Hindley by NERC grant NE/S00985X/1.

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

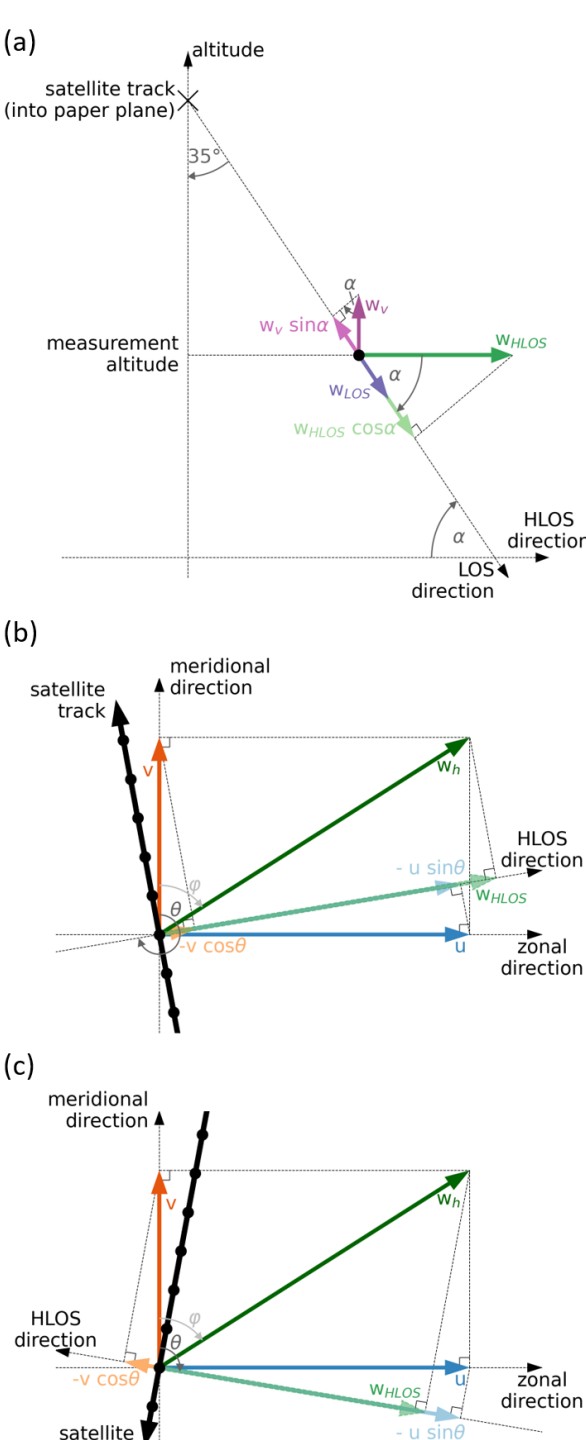


**Figure 1: Measurement geometry of Aeolus. Panel (a) is in the vertical along-line-of-sight plane and panels (b) and (c) show the horizontal plane for ascending and descending orbits, respectively, for typical track angles of 10° from North. For better illustration, panel (a) shows a situation with extraordinary strong vertical wind $w_v$.**



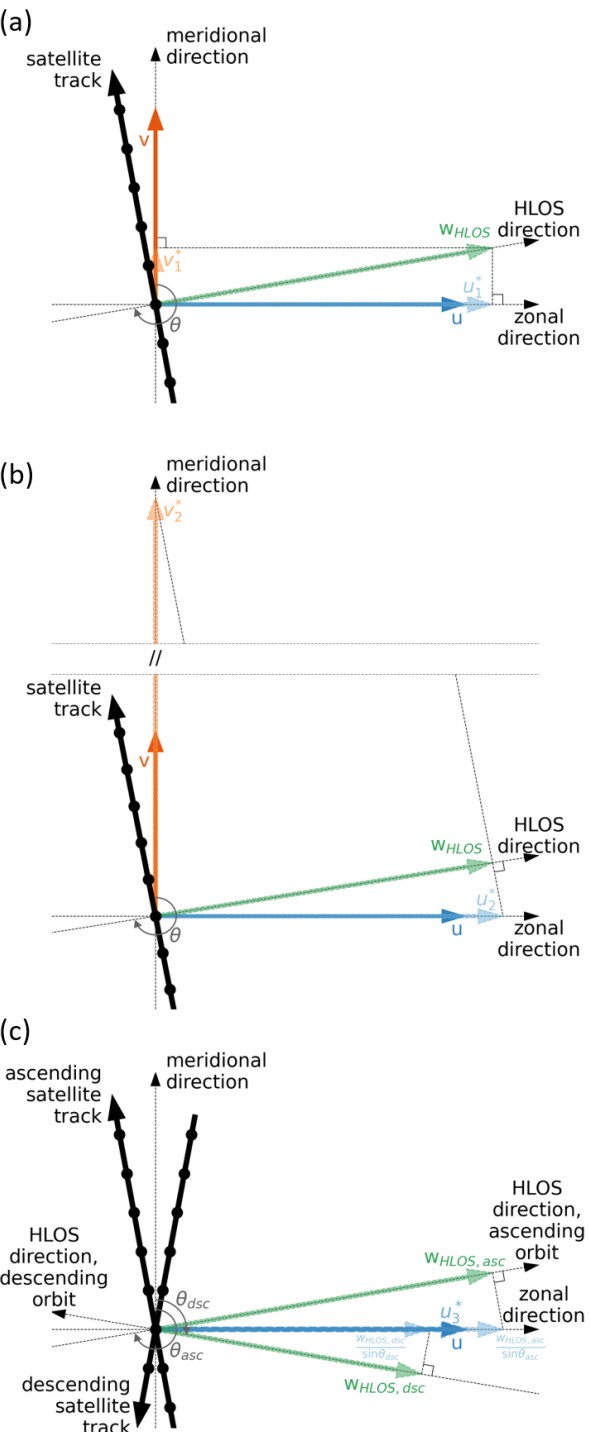

**Figure 2: Geometric concepts of the three methods to convert Aeolus $w_{HLOS}$ to zonal and meridional wind, which are introduced in this paper. Panel (a) shows the projection of $w_{HLOS}$ onto zonal and meridional axes (Method 1), panel (b) the assumption of zero zonal or meridional wind (Method 2) and panel (c) the combination of ascending and descending measurements (Method 3).**

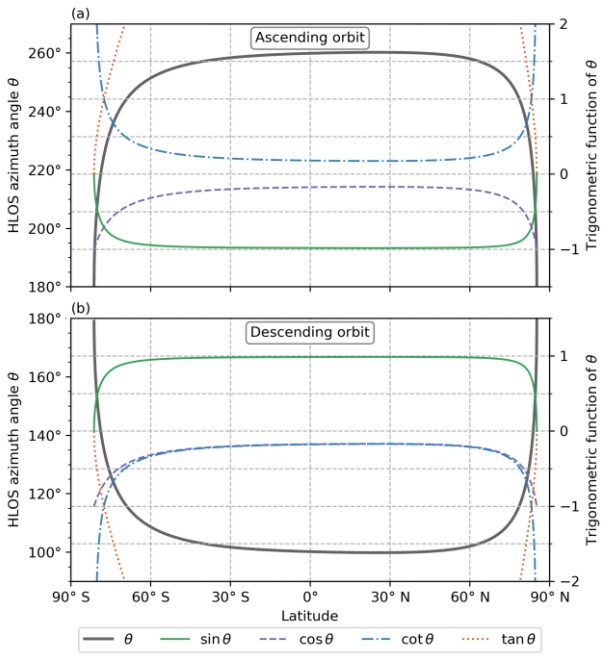

**Figure 3: Aeolus azimuth angle and its trigonometric functions with respect to latitude for ascending (a) and descending (b) orbits.**


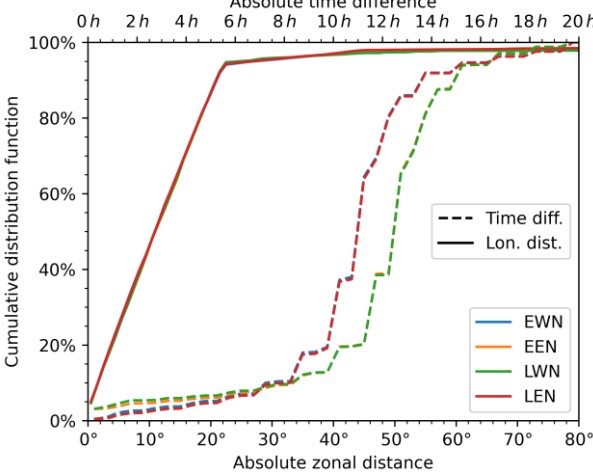

**Figure 4: Spatial and temporal distance between Aeolus measurement location and the corresponding four nearest neighbours EWN, EWN, LWN, and LEN for all available data from January 2021.**




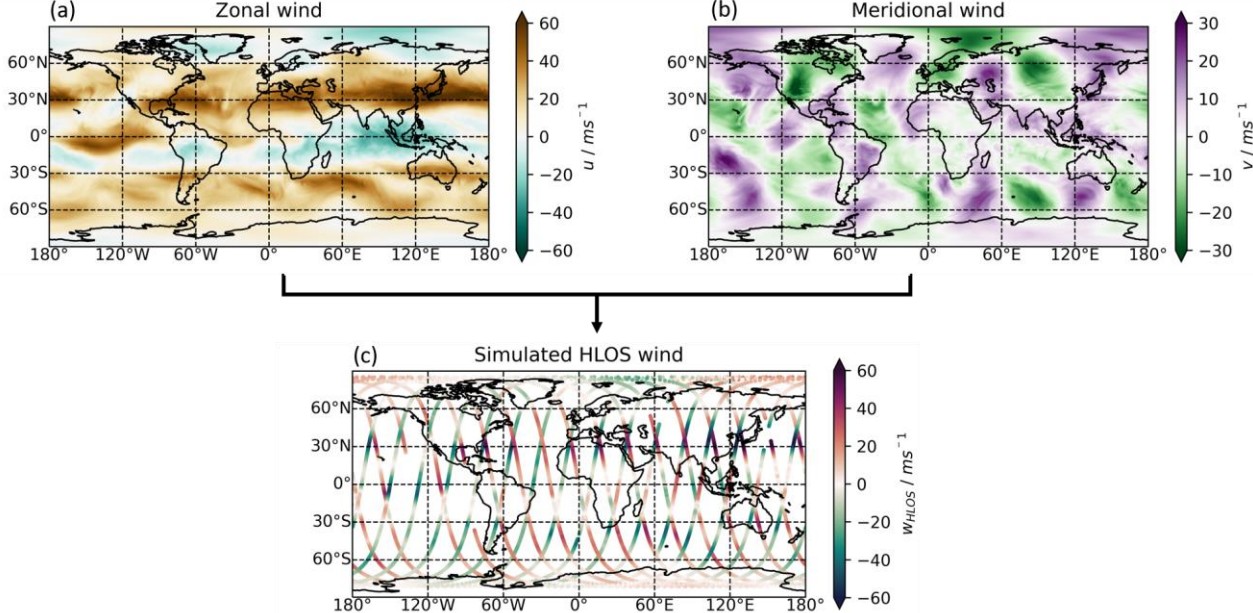

**Figure 5: Simulation of Aeolus HLOS wind from ERA5 model data. Panels (a) and (b) show the zonal and meridional wind fields from ERA5 on 15 January 2021, 12 UTC for an altitude of 15 km. Panel (c) shows the simulated Aeolus HLOS wind along track on 15 January 2021. This simulated Aeolus HLOS wind is constructed through trilinear interpolation of hourly ERA5 zonal and meridional wind fields onto the exact times and locations of real Aeolus observations, followed by a conversion into simulated Aeolus HLOS wind following Eq. (3).**






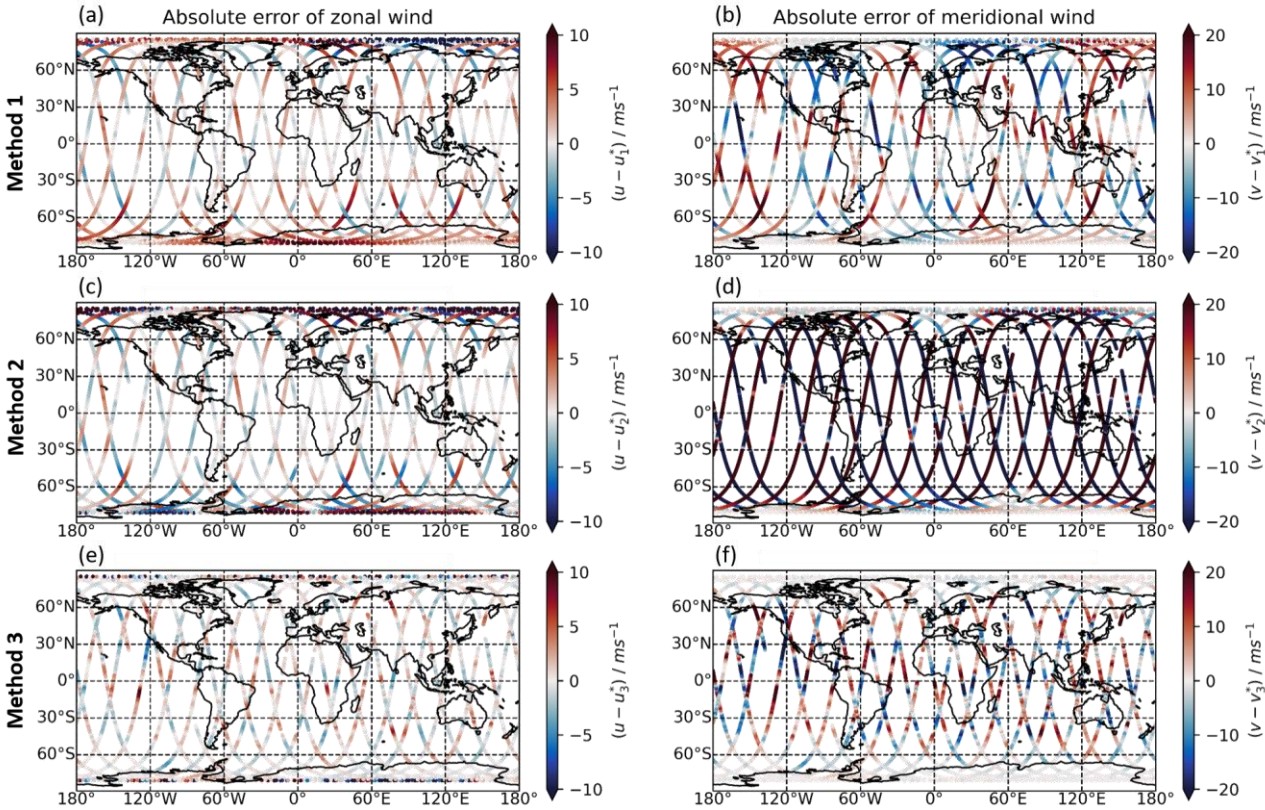

**Figure 6: Maps of absolute errors of the different methods for zonal (left) and meridional (right) wind on 15 January 2021.**




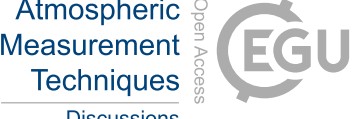

**Figure 7: Statistical distribution of absolute and relative errors for the different methods versus latitude (x-axes). Data from the whole month of January 2021 is used here.**



**Figure 8: Influence of spatial and temporal interpolation on the total error statistics of Method 3. Panels a & d are a copy of Figure 7 c & i. The first two rows show the error statistics for zonal and meridional wind. In the third row, the spatial (g) and temporal (h) distance to the four nearest neighbours used for interpolation is plotted. The bottom row depicts the mean difference between maximal and minimal wind speed within 48° longitude (i) and 24 h (j) derived from ERA5 data. Data from the whole month of January is used for this variability analysis.**







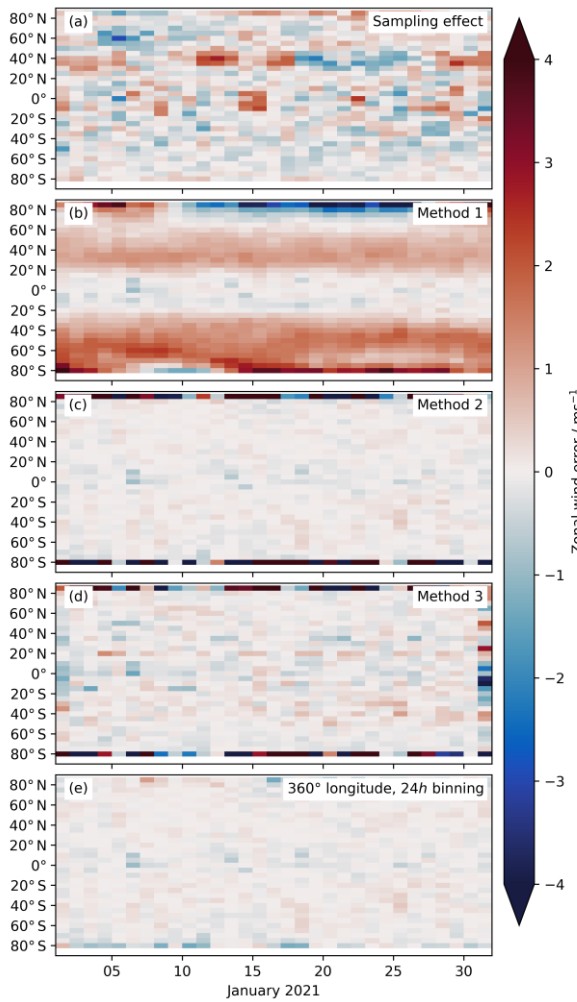

**Figure 9: Absolute errors in the daily zonal mean zonal wind calculation for January 2021. Panel (a) shows errors originating in the Aeolus sampling, panels (b-d) the errors when using the three conversion methods, respectively, and panel (e) the errors for Method 3, but with a slightly different colocation/binning algorithm. The sampling bias shown in Panel (a) has been removed from the error estimates in panels (b-e).**
