# Peer review of "On the derivation of zonal and meridional wind components from Aeolus horizontal line-of-sight wind"

_Atmospheric Measurement Techniques, 2021_

## Author Response (AR3)

**On the derivation of zonal and meridional wind components from Aeolus horizontal line-of-sight wind**

Isabell Krisch[1], Neil P. Hindley[2], Oliver Reitebuch[1], and Corwin J. Wright[2]

[1]Deutsches Zentrum für Luft- und Raumfahrt e.V. (DLR), Institut für Physik der Atmosphäre, Oberpfaffenhofen, Germany
[2]Centre for Space, Atmospheric and Oceanic Science, University of Bath, UK

*Correspondence to*: Isabell Krisch (isabell.krisch@dlr.de)

**Response to Anonymous Referee #1**

Dear Anonymous Referee #1,

Thank you very much for reviewing our manuscript and for your helpful comments. We added a section to discuss the impact of our methods on possible future Doppler-Wind-Lidar scenarios and applied some minor changes to the manuscript. Please find detailed answers on all your comments below.

Sincerely,

Isabell Krisch on behalf of all Co-Authors

1. **Eq. 11 is expressed in a slightly different way as Eq. 10 in the first line. It's better to rewrite the Eq.11 for consistency.**

   Both equations (10 & 11) were slightly amended for consistency, as also proposed by Anonymous Referee #2:

$$
\begin{aligned}
u_3^* &= -0.5 \cdot \left( \frac{w_{HLOS,asc}}{\sin \theta_{asc}} + \frac{w_{HLOS,dsc}}{\sin \theta_{dsc}} \right) \\
&= -0.5 \cdot \left( \frac{-u_{asc} \sin \theta_{asc} - v_{asc} \cos \theta_{asc}}{\sin \theta_{asc}} + \frac{-u_{dsc} \sin \theta_{dsc} - v_{dsc} \cos \theta_{dsc}}{\sin \theta_{dsc}} \right) \\
&= 0.5 \cdot (u_{asc} + v_{asc} \cot \theta_{asc} + u_{dsc} + v_{dsc} \cot \theta_{dsc}) \\
&= 0.5 \cdot (u_{asc} + u_{dsc}) + 0.5 \cdot \cot \theta_{asc} \cdot (v_{asc} - v_{dsc}),
\end{aligned}
\tag{10}
$$

$$
\begin{aligned}
v_3^* &= -0.5 \cdot \left( \frac{w_{HLOS,asc}}{\cos \theta_{asc}} + \frac{w_{HLOS,dsc}}{\cos \theta_{dsc}} \right) \\
&= -0.5 \cdot \left( \frac{-u_{asc} \sin \theta_{asc} - v_{asc} \cos \theta_{asc}}{\cos \theta_{asc}} + \frac{-u_{dsc} \sin \theta_{dsc} - v_{dsc} \cos \theta_{dsc}}{\cos \theta_{dsc}} \right) \\
&= 0.5 \cdot (u_{asc} \tan \theta_{asc} + v_{asc} + u_{dsc} \tan \theta_{dsc} + v_{dsc}) \\
&= 0.5 \cdot \tan \theta_{asc} \cdot (u_{asc} - u_{dsc}) + 0.5 \cdot (v_{asc} + v_{dsc}).
\end{aligned}
\tag{11}
$$

2. All three methods produce reliable zonal wind estimates between 70° S and 70° N with absolute errors typically below 5 ms[-1]. Method 3 is the only method able to produce reliable meridional winds at all latitudes. It's straightforward that the error of Method 1 and Method 2 depends on how well the zonal and meridional wind components is projected onto Aeolus Line-of-sight measurement. It's a latitude related error different from the equator to the poles. Method 3 is based on the combination of two measurements in the collocated analysis region, the error of which relies on temporal and spatial interpolation. This method can be analogous to the velocity-azimuth processing technique, so called VAP method for single weather radar and wind lidar. The colocation analysis would be instructive for future Aeolus follow-on mission, for instance the two-satellite constellation to provide two independent measurements for zonal and meridional wind components. It would be great if authors can comment on that two points above.

Yes, method 3 is inspired by the VAP or more commonly VAD (velocity azimuth display) method. A note on this has been added to the manuscript:

*The third method is inspired by the velocity–azimuth display (VAD) technique for single ground-based or airborne radar or lidar instruments (e.g. Browning and Wexler, 1968; Reitebuch et al. 2001; Witschas et al., 2017): The laser or radar beam is actively steered in different azimuth directions to retrieve a horizontal wind vector by combining different LOS measurements. Aeolus cannot steer its LOS, but we can use the geometrical differences between ascending and descending orbits and combines measurements from both to more accurately estimate the true zonal and meridional wind over a specific region.*

Regarding possible Aeolus follow-on scenarios and additional section has been added to the manuscript briefly touching this issue:

**Impact of possible future Doppler-Wind-Lidar scenarios on the accuracy of Method 3**

*Although a detailed discussion of possible future Doppler-Wind-Lidar (DWL) scenarios (e.g. Marseille et al., 2008; Baker et al., 2014) is beyond the scope of this paper, we would like to briefly comment here on the impact of dual-perspective and multiple satellite constellation scenarios on the accuracy of derived winds from our Method 3.*
*A dual-perspective DWL would provide two LOS wind measurements under different azimuth angles from one satellite (e.g. Baker et al., 2014, their Fig. 12). This would be ideal, because the time difference and spatial distance between these two wind measurements would be negligible and the systematic errors of our Method 3 would become very small.*
*Another scenario discussed for a future DWL mission is a multi-satellite constellation. In this scenario, the accuracy of our Method 3 strongly depends on two key characteristics of such a constellation: how far apart in time and space are the two (or more) satellites, and do the different instruments have the same LOS with respect to flight-direction? In a constellation with two identical satellites that both have the same LOS direction in the same orbit plane and only a small shift in time and space (e.g. Tandem-Aeolus scenario of Marseille et al., 2008), errors in our Method 3 would only be slightly reduced compared to a single satellite constellation. This is because although the spatial distance between the nearest neighbours would decrease by a factor of two (or more, for more satellites) in such a constellation due to the shift in orbit, the time difference between ascending and descending measurements would remain large. However, if the tandem constellation described above was amended such that one of the satellites had a different LOS viewing direction, errors in our derived winds would be strongly reduced and their reliability greatly increased. This is because, in addition to the close spatial separation of the different LOS measurements, there would only be a small time difference.*
*Thus, for deriving the zonal and meridional winds from spaceborne DWL measurements, a dual-perspective DWL would perform best, followed by a multiple satellite constellation with differing LOS. A multiple satellite constellation with similar LOS for all satellites is expected to only slightly improve the derivation of zonal and meridional wind components compared to Aeolus.*
*In contrast, for NWP use, it is more important to get a high geographical coverage of wind profiles (e.g. multiple satellite constellation) than measuring in dual perspective (Marseille et al., 2008).*

**Response to Anonymous Referee #2**

Dear Anonymous Referee #2,

Thank you very much for reviewing our manuscript and for your helpful comments. We applied some minor changes to the manuscript as proposed. Additionally, we added a section to discuss the impact of our methods on possible future Doppler-Wind-Lidar scenarios, as it was requested by Referee #1. Please find detailed answers on all your comments and the text of the new section below.

Sincerely,

Isabell Krisch on behalf of all Co-Authors

1. **Sect. 3.1: For Method #1 it should be clarified that this method assumes that the wind direction and the LOS direction are always "accidentally" the same. For testing a simple method to derive wind vectors, this assumption makes absolutely sense, but you should mention that strictly speaking this is not a physically well-reasoned assumption.**

    The authors completely agree with the referee on this topic, but some scientific studies (preprints) have used this method, so we thought it was important to be included. We added a more detailed explanation on this topic to the manuscript:

    *This very simple approach is nothing else than assuming that the horizontal wind direction is aligned with HLOS, which is not a physically well-reasoned assumption. Nevertheless, it is already used in the community (e.g. preprint of Wright et al., 2021, and Chou et al., 2021) and we will show later that under certain conditions it provides reasonable estimates for the zonal wind.*

2. **l.182: Why did you select 20hr of miss-time? Is there a reason?**

    20hrs of miss-time was chosen to make sure the nearest neighbours in time (which are due to the orbit geometry around 10-14hrs away) are included, but at the same time to reduce the data amount for calculation as much as possible. For better reasoning this miss-time has been increased to 24hrs now (sun-synchronous orbit geometry of Aeolus). The results did not change. An explanation has been added to the manuscript:

    *Twenty-four hours of miss-time are chosen due to the sun-synchronous orbit geometry of Aeolus.*

3. **l.186: Interpolation in time is also linear?**

    Yes, the interpolation in time is also linear. This has been clarified in the manuscript.

4. **Fig.4: Please clarify: This figure combines data from all latitudes? Why do the lon. dist. curves in Fig.4 not show steps, similar to the time diff. curves? Would the statistics look quite different if only a limited latitude range is considered?**

    Yes, this figure combines data from all latitudes. The longitudinal distance curves would show steps when looking at single latitudes only. When looking at Fig. 5c, it becomes obvious why no steps are observed when all latitude bands are included: The distances between the ascending and descending orbits is smoothly decreasing and increasing again when following one track from north to south or vice versa. For the temporal distance this is different: One orbit takes

around 1.5h and during half of the orbit, the satellite is on the other side of the Earth (= in the same orbit phase as the comparison orbit), before reappearing again. During this time period, no collocated measurements are counted and thus the step. This has been clarified in the manuscript:

*For this figure, all latitudes are used. This leads to the continuous transition of the spatial distance from 0% to 100%. When looking at single latitudes this transition would be step-wise, but the overall evolution would be similar. For the temporal distance this is different: One orbit takes around 1.5h and during half of the orbit, the satellite is on the other side of the Earth, before reappearing again. During these roughly 45min no collocated measurements are acquired, which explains the steps in the temporal distance distribution. For a single point / latitude these steps would be more emphasised, but, again, the overall evolution would not change.*

5. **l.199: What are "Rayleigh clear measurement locations"? Do you mean cloud-free, or locations where Rayleigh wind observations of Aeolus show only small errors?**

Yes, only cloud-free Rayleigh measurements are used. Within the Aeolus L2B products a flag exists to filter for "cloudy" or "clear" measurements. This is why we used the word "clear" in the text as well. No additional filtering on the product quality (e.g. error estimates) is used, as the measurement points are anyway populated with synthetic measurement data from ERA5.

6. **Sect.6: Error estimation is only performed for the month of January. As the error depends on the angle between the real wind and the Aeolus LOS, do you think that error estimates will be significantly different for other months/seasons?**

The error estimation has been performed also for other months (March, June, October) with no significant differences in the results. For method 1, the mean bias slightly changes its structure with respect to latitude with the season, but no change in bias magnitude is observed. A sentence explaining this has been added to the manuscript:

*Processing of data from other months (not shown) did not lead to significant differences in the results are discussed compared to the January 2021 dataset. Thus, in the following, the results for zonal and meridional wind are discussed only for the January 2021 dataset.*

7. The following section has been added to the manuscript:

*Impact of possible future Doppler-Wind-Lidar scenarios on the accuracy of Method 3*

*Although a detailed discussion of possible future Doppler-Wind-Lidar (DWL) scenarios (e.g. Marseille et al., 2008; Baker et al., 2014) is beyond the scope of this paper, we would like to briefly comment here on the impact of dual-perspective and multiple satellite constellation scenarios on the accuracy of derived winds from our Method 3.*
*A dual-perspective DWL would provide two LOS wind measurements under different azimuth angles from one satellite (e.g. Baker et al., 2014, their Fig. 12). This would be ideal, because the time difference and spatial distance between these two wind measurements would be negligible and the systematic errors of our Method 3 would become very small.*
*Another scenario discussed for a future DWL mission is a multi-satellite constellation. In this scenario, the accuracy of our Method 3 strongly depends on two key characteristics of such a constellation: how far apart in time and space are the two (or more) satellites, and do the different instruments have the same LOS with respect to flight-direction? In a constellation with two identical satellites that both have the same LOS direction in the same orbit plane and only a small shift in time and space (e.g. Tandem-Aeolus scenario of Marseille et al., 2008), errors in our Method 3 would only be slightly reduced compared to a single satellite constellation. This is because although the spatial distance between the nearest neighbours would decrease by a factor of two (or more, for more satellites) in such a constellation due to the shift in orbit, the time difference between ascending and descending measurements would remain large. However, if the tandem constellation described above was amended such that one of the satellites had a different LOS viewing direction, errors in our derived winds would be strongly reduced and their reliability greatly increased. This*

*is because, in addition to the close spatial separation of the different LOS measurements, there would only be a small
time difference.*

145    *Thus, for deriving the zonal and meridional winds from spaceborne DWL measurements, a dual-perspective DWL
would perform best, followed by a multiple satellite constellation with differing LOS. A multiple satellite constellation
with similar LOS for all satellites is expected to only slightly improve the derivation of zonal and meridional wind
components compared to Aeolus.*

   *In contrast, for NWP use, it is more important to get a high geographical coverage of wind profiles (e.g. multiple*
150    *satellite constellation) than measuring in dual perspective (Marseille et al., 2008).*

**Response to Anonymous Referee #3**

Dear Anonymous Referee #3,

Thank you very much for reviewing our manuscript and for your helpful comments. Following your suggestions, we applied
155 some minor changes to the manuscript. Please find detailed answers on all your comments below.

Sincerely,

Isabell Krisch on behalf of all Co-Authors

1. **Remove +/- on line 18, to be consistent with line 15.**

    Changed in the manuscript.
160

2. **Line 40: "and the error estimates too high to properly resolve even strong vertical winds". Can you provide
    a reference showing this? Or by logic reasoning? Please add to the text.**

    Some more details on the averaging lengths and the error estimates are added to the manuscript. Also this topic
    regarding the vertical wind is also discussed in more detail at the end of Section 2.

165    *The vertical wind component will not be considered here, because the typical horizontal and vertical averaging
    lengths of Aeolus measurements (90 km and 0.5 – 2 km, respectively) are too coarse and the error estimates too
    high (usually between 3 and 7 ms$^{-1}$; Rennie et al., 2021) to properly resolve even strong vertical winds, e.g. in the
    atmospheric boundary layer, within convection, or in gravity waves.*

170 3. **Line 52: "over a long time periods". Remove "a".**

    Changed in the manuscript.

4. **Caption figure 7. Please explain the green and red dashed lines in the caption.**

    Explanation added to the manuscript:
175    *The red dashed lines indicate zero absolute and relative wind errors, the green dashed lines in the relative error
    plots indicate errors of 100%.*

5. **Line 204: methods -> method**

    Changed in the manuscript.
180

6. **Line 223: "At a first glance, 5 ms⁻¹ might seem high." I would say, a systematic error of 5 m/s IS high. You cannot simply compare systematic with random error to suggest the opposite. A similar method applied to Aeolus-2, with expected much smaller random error, would invalidate this statement. Please add this nuance in the text.**

As Figure 7 shows, the 5 ms$^{-1}$ error in the zonal wind is mostly a randomly distributed error and not a constant bias. Only Method 1, shows some kind of constant bias depending on latitude. Thus, the 5 ms$^{-1}$ error has to be compared to the random error of Aeolus and not the systematic bias. Regarding Aeolus-2 and the overall large biases, the manuscript has been amended as follows:

*At a first glance, 5 ms$^{-1}$ might seem high. However, one should keep in mind that Aeolus is not designed to measure the zonal and / or meridional wind component directly. Thus, such larger errors, unfortunately, have to be expected. Additionally, these errors are, at least for the current Aeolus instrument, in a similar range as the typical random errors of Aeolus HLOS wind measurements in the UTLS (usually between 3 and 7 ms$^{-1}$; Rennie et al., 2021).*

**Response to Editor**

Dear Editor,

Thank you very much for reviewing our manuscript and for your helpful comments.

I would like to point out here that the intended target of this work is the atmospheric dynamics research community (and not the NWP one). This is also clearly stated in the introduction of the manuscript:

*The present paper will investigate if and how a conversion to zonal and meridional wind is possible using solely Aeolus measurements without requiring additional information (for example from NWP models). The presented methods are especially useful for studying atmospheric phenomena over long time periods and large geographic areas (e.g. zonal mean values), where larger Aeolus observation samples can be used. The aim of this paper is to provide scientists studying synoptic scale phenomena especially in the stratosphere (e.g. SSWs, QBO) with a toolbox to convert the Aeolus L2B products into zonal and meridional wind components keeping the limitations of such conversion methods in mind.*

All methods presented in this manuscript are purely measurement based and do not require any model information. This is the main strength of these techniques and was a conscious decision by the authors. The only model data used in the manuscript is ERA5 data, which is used solely for concept validation purpose.

The goal of this work is not to discuss the accuracy of measurement-based vs model-based results, but to address the different methods to process Aeolus measurements in order to derive zonal and meridional wind components. In Wright et al. (2021) and Banyard et al. (2021), for instance, such purely measurement-based approaches are already in use.

Following your suggestions, we applied some minor changes to the manuscript. Please find detailed answers on all your comments below.

Sincerely,

Isabell Krisch on behalf of all Co-Authors

1. **L 365: Aeolus data are now used in advanced data assimilation systems with appropriate temporal and spatial interpolation in 4Dvar. The resulting winds have accuracy of about 2 m/s, both zonally and meridionally. Furthermore, the temporal evolution and change of these winds between ascending and descending passes is well captured in the circulation model used. As I understand it, this is exploited now in the manuscript by subtracting the sampling error derived from the ECMWF model prior winds, is that right? Would you then not recommend to interpolate the increment vector components with respect to the ECMWF model, rather than advance an inferior data analysis system for the full vector winds?**

   Please see general response to comments by the editor above.

2. **Comment to the authors response: I miss some of the suggested changes in the manuscript?**

   Thank you for checking this. You are absolutely right, we missed the inclusion of the proposed changes for comment #5 of Anonymous Referee #3. This has now finally been added to the manuscript.

   All previous provided track changes manuscripts were always based on the previous version (e.g. after first revision, …). To avoid confusion, we changed this now and the manuscript including all track changes is now based on the first manuscript submitted to the journal (e.g. all changes applied during the review process are highlighted now).

3. **L 33&34 of authors response: Rather than "stirred", presumably you mean "steered"?**

   This was corrected in the authors response and the manuscript.

4. **L 45 of authors response: Is a similar effect not achieved by working with increment vector (HLOS) components, rather than full vectors (see also comment above)? Presumably, in that case most temporal and diurnal effects are taken out as they are well represented in the ECMWF reference fields temporal evolution.**

   Please see general response to comments by the editor. The main strength of all tested methods is that they do not rely on model information. Using increment vectors would be in contradiction with the purpose of this work.

5. **L 52 of authors response: What is a small shift in time and space? Marseille et al. show that 45 minutes and 11 degrees longitude between 2 subsequent orbits is more than sufficient to provide independent wind information to a NWP model. This is in line with the (Aeolus mission) idea that temperature information from satellites is sufficient to initialize the large balanced scales with 2D turbulence, while the 3D information from Aeolus is particularly helpful to initialize 3D turbulence on the smaller scales (500 km). Like in Marseille et al. one would expect that increased sampling resolution will bring relevant new information in these methods too? The time difference can be addressed by correcting for the sampling error or by working with Aeolus analysis increments I guess. Besides, it appears that in addition to the mean (HLOS vector component) increments, also the variance of the increments would be useful to locally analyse for ascending and descending orbits?**

   The 45 minutes and 11 degrees longitude shift between 2 subsequent orbits proposed by Marseille et al. would only slightly reduce the errors of Method 3, because although the spatial distance between the nearest neighbours would decrease by a factor of two (or more, for more satellites) in such a constellation due to the shift in orbit, the time difference between ascending and descending orbits would remain large.

   This explanation is already contained in the manuscript in Section 6.4, but for further clarification, "*between ascending and descending measurements*" was added to the respective paragraph.

   Please see general response to comments by the editor regarding the part on NWP.

6. **L 61 of authors response: The paragraph that starts here is rather imprecise in a broader context and prone to misinterpretation I feel. First, several comprehensive studies have been performed, looking for those DWL configurations that bring most new information to meteorological analyses, like Marseille et al.. The latter concludes that multiple Aeolus satellites appear to perform best, at least outside the tropics. It is supported by the idea of the different turbulence regimes as outlined above. Then, second, the findings with the simplified analyses methods presented here do not appear to follow this finding. In my view, the first question would be: why not? The second question would be: are these simplified methods then useful at all? Or should we be looking at mean analysis increments and their variability, as these better depict what information really is missing from comprehensive meteorological analyses? I would completely reconsider the text on Aeolus follow-on with the above broader context in mind.**

Please see general response to comments by the editor above.

In contrast to targeting the atmospheric dynamics research community, Marseille et al. focus on the use of Doppler-Wind-Lidar data for NWP. Given the different use cases and focus, it is not unreasonable to come to different conclusions. In addition, the discussion on Aeolus follow-on, which the authors originally did not intend to be part of this paper and later added due to the comments from reviewer #1, is not meant to be exhaustive.

For clarity, the following sentence has been added to the paragraph on possible future Doppler-Wind-Lidars to make the different use cases of Marseille et al. and the present manuscript more visible and to remove any possible contradiction:

*In contrast, for NWP use, it is more important to get a high geographical coverage of wind profiles (e.g. multiple satellite constellation) than measuring in dual perspective (Marseille et al., 2008).*

**References**

Baker, W. E., Atlas, R., Cardinali, C., Clement, A., Emmitt, G. D., Gentry, B. M., Hardesty, R. M., Källén, E., Kavaya, M. J., Langland, R., Ma, Z., Masutani, M., McCarty, W., Pierce, R. B., Pu, Z., Riishojgaard, L. P., Ryan, J., Tucker, S., Weissmann, M., and Yoe, J. G.: Lidar-Measured Wind Profiles: The Missing Link in the Global Observing System, Bulletin of the American Meteorological Society, 95(4), 543-564, https://doi.org/10.1175/BAMS-D-12-00164.1, 2014.

Banyard, T., Wright, C., Hindley, N., Halloran, G., and Osprey, S.: The 2019/2020 QBO Disruption in ADM-Aeolus Wind Lidar Observations, EGU General Assembly 2021, online, 19–30 Apr 2021, EGU21-16107, https://doi.org/10.5194/egusphere-egu21-16107, 2021.

Browning, K., and Wexler, R.: The determination of kinematic properties of a wind field using Doppler radar. J. Appl. Meteor., 7, 105–113, https://doi.org/10.1175/1520-0450(1968)007<0105:TDOKPO>2.0.CO;2, 1968.

Chou, C.-C., Kushner, P. J., Laroche, S., Mariani, Z., Rodriguez, P., Melo, S., and Fletcher, C. G.: Validation of the Aeolus Level-2B wind product over Northern Canada and the Arctic, Atmos. Meas. Tech. Discuss. [preprint], https://doi.org/10.5194/amt-2021-247, in review, 2021.

Marseille, G.-J., Stoffelen, and A., Barkmeijer, J.: Impact assessment of prospective spaceborne Doppler wind lidar observation scenarios, Tellus A: Dynamic Meteorology and Oceanography, 60:2, 234-248, https://doi.org/10.1111/j.1600-0870.2007.00289.x, 2008.

Reitebuch, O., Werner, Ch., Leike, I., Delville, P., Flamant, P. H., Cress, A., and Engelbart, D.: Experimental Validation of Wind Profiling Performed by the Airborne 10-µm Heterodyne Doppler Lidar WIND. J. Atmos. Ocean. Tech. 18, 1331-1344,

295 https://doi.org/10.1175/1520-0426(2001)018%3C1331:EVOWPP%3E2.0.CO;2, 2001.

Witschas, B., Rahm, S., Dörnbrack, A., Wagner, J., and Rapp, M.: Airborne Wind Lidar Measurements of Vertical and Horizontal Winds for the Investigation of Orographically Induced Gravity Waves, Journal of Atmospheric and Oceanic Technology, 34(6), 1371-1386, https://doi.org/10.1175/JTECH-D-17-0021.1, 2017.

Wright, C. J., Hall, R. J., Banyard, T. P., Hindley, N. P., Krisch, I., Mitchell, D. M., and Seviour, W. J. M.: Dynamical and

300 surface impacts of the January 2021 sudden stratospheric warming in novel Aeolus wind observations, MLS and ERA5, Weather Clim. Dynam., 2, 1283–1301, https://doi.org/10.5194/wcd-2-1283-2021, 2021.